Icaritin protects SH-SY5Y cells transfected with TDP-43 by alleviating mitochondrial damage and oxidative stress

Zhou Yongjian 1 2
Huang Nanqu 3
Li Yuanyuan 3
Ba Zhisheng 3
Zhou Yanjun 2
Luo Yong luoyong@zmu.edu.cn 2
1 Department of Neurology, Xiangtan Central Hospital , Xiangtan , Hunan , China
2 Department of Neurology, Third Affiliated Hospital of Zunyi Medical University (The First People’s Hospital of Zunyi) , Zunyi , Guizhou , China
3 National Drug Clinical Trial Institution, Third Affiliated Hospital of Zunyi Medical University (The First People’s Hospital of Zunyi) , Zunyi , Guizhou , China
Stochaj Ursula
Electronic publication date: 2021 Aug 10
Publication date: 2021
Volume: 9
Electronic Location ID: e11978
Received 2021 Apr 2; Accepted 2021 Jul 24
Copyright: ©2021 Zhou et al.
Copyright year: 2021
Copyright holder: Zhou et al.
License: This is an open access article distributed under the terms of the Creative Commons Attribution License, which permits unrestricted use, distribution, reproduction and adaptation in any medium and for any purpose provided that it is properly attributed. For attribution, the original author(s), title, publication source (PeerJ) and either DOI or URL of the article must be cited.
License URL: https://creativecommons.org/licenses/by/4.0/

Keywords: Icaritin, TDP-43, Mitochondria, Oxidative stress, SH-SY5Y cells

Funding: National Natural Science Foundation of China 81860710 Funds of Zunyi Science and Technology Bureau 2017-29 2020-107 Science and Technology Fund of Guizhou Provincial Health Commission gzwjkj2019-1-064 This work was supported by the National Natural Science Foundation of China (No. 81860710), the Funds of the Zunyi Science and the Technology Bureau (No.2017-29, 2020-107) and the Science and Technology Fund of Guizhou Provincial Health Commission (No. gzwjkj2019-1-064). The funders had no role in study design, data collection and analysis, decision to publish, or preparation of the manuscript.

==============================
Background

The aim of this study was to investigate the effect of icaritin (ICT) on TAR DNA-binding protein 43 (TDP-43)-induced neuroblastoma (SH-SY5Y) cell damage and to further explore its underlying mechanisms.

Methods

To investigate the possible mechanism, TDP-43 was used to induce SH-SY5Y cell injury. Cell viability was evaluated by the CCK-8 assay. The mitochondrial membrane potential (MMP) was determined with JC-1. The expression levels of TDP-43 and cytochrome C (CytC) were measuring by Western blotting. Changes in adenosine 5′-triphosphate (ATP) content, total antioxidative capacity (T-AOC), glutathione peroxidase (GSH-Px) activity, superoxide dismutase (SOD) activity and malondialdehyde (MDA) content were detected with specific kits.

Results

The results showed that ICT reduced the cell damage induced by TDP-43. ICT reduced the expression level of TDP-43; increased ATP content and the MMP; decreased CytC expression; increased T-AOC and GSH-Px, total SOD (T-SOD), copper/zinc SOD (CuZn-SOD), and manganese SOD (Mn-SOD) activity; and decreased MDA content.

Conclusions

The results suggest that ICT has a protective effect on TDP-43-transfected SH-SY5Y cells that is related to reductions in TDP-43 expression and mitochondrial damage and alleviation of oxidative stress.

Introduction

TAR DNA-binding protein 43 (TDP-43) is a multifunctional nucleic acid-binding protein composed of 414 amino acids and six exons. Under physiological conditions, it can regulate RNA splicing, transport and translation in a variety of ways and is involved in the biosynthesis of microRNAs. It is an important transcriptional regulator (Sephton et al., 2012). This protein was discovered by researchers in 1995 (Ou et al., 1995). In addition to having normal physiological functions, TDP-43 is closely related to amyotrophic lateral sclerosis (ALS), frontotemporal lobar dementia (FTLD), Alzheimer’s disease (AD), Parkinson’s disease (PD) and other neurodegenerative diseases (Gao et al., 2019). Mitochondria are organelles with a double membrane structure that are involved in a variety of important biological functions and are the main sites of energy production and aerobic respiration in cells. Adenosine 5′-triphosphate (ATP) produced by mitochondria is the main energy source for various nerve cell activities, such as nerve conduction, synaptic transmission and axonal transport (Cavallucci, Ferraina & D’Amelio, 2013). Energy metabolism disorder in the brain not only affects the functional state of the brain and mitochondrial function but also reduces the antioxidant capacity of mitochondria, leading to oxidative stress (Islam, 2017; Nissanka & Moraes, 2018). Increased TDP-43 expression can cause mitochondrial dysfunction, including a decreased mitochondrial membrane potential (MMP), resulting in abnormal membrane permeability and thus increased cytochrome C (CytC) expression in the cytoplasm. Overexpression of TDP-43 can inhibit mitochondrial activity and reduce mitochondrial ATP synthesis (Wang et al., 2019). TDP-43 can also lead to an imbalance between oxidation and antioxidation, resulting in a decrease in total antioxidant capacity (T-AOC) and an imbalance in oxidation- and antioxidant-related enzymes such as glutathione peroxidase (GSH-Px), superoxide dismutase (SOD) and malondialdehyde (MDA), leading to oxidative stress (Wang et al., 2019). A large number of studies have shown that mitochondrial damage and oxidative stress induced by TDP-43 play an important role in TDP-43 proteinopathy (Chang et al., 2016; Wang et al., 2019). Preventing abnormal aggregation and localization of TDP-43 in cells effectively prevents mitochondrial damage, alleviates oxidative stress, prevents degeneration of neurons, and thus prevents the occurrence and development of neurodegenerative diseases (Wang et al., 2016).

Icaritin (ICT), a component of Epimedium, is rich in flavonoids and easily crosses the blood–brain barrier (Li et al., 2020b). It improves learning and memory, fights inflammation, scavenges oxygen free radicals, slows aging and protects neuronal function (Feng et al., 2019; Hao et al., 2019; Wang et al., 2007). Studies have shown that ICT can protect neurons by ameliorating mitochondrial damage and oxidative stress (Wu et al., 2021; Xu et al., 2021). Therefore, we hypothesized that ICT can alleviate energy metabolism disorder in the brain and that its mechanism might be related to a reduction in TDP-43 expression.

Therefore, to confirm our hypothesis, the protective effect of ICT against TDP-43-induced mitochondrial damage in SH-SY5Y cells (neurons originated from neuroblastoma cells, a common nerve cell line) (Agholme et al., 2010) was evaluated, and the possible mechanism was discussed.

Materials and Methods

Reagents

Reagent-grade ICT, (purity 99.5% by HPLC analysis) was obtained from Zhongke Quality Inspection Biotechnology Co., Ltd. (Beijing, China) and dissolved in dimethyl sulfoxide (DMSO); the final concentration of DMSO in the medium was less than 0.1% (v/v). Human TDP-43 and polybrene were purchased from Hanbio Biotechnology Co., Ltd. (Shanghai, China), a TransZol Up Plus RNA Kit and EasyScript One-Step gDNA Removal and cDNA Synthesis SuperMix were purchased from TransGen Biotechnology Co., Ltd. (Beijing, China) and NuHi Robustic SYBR Green Mix was purchased from Nuhigh Biotechnologies Co., Ltd. (Suzhou, China). High-Sig ECL Western Blotting Substrate was purchased from Tanon Technology Co., Ltd. (Shanghai, China). RIPA buffer (high) (R0010) was purchased from Solarbio Life Science (Beijing, China). A GAPDH antibody, a TDP-43 antibody, a CytC antibody, and HRP-conjugated Affinipure goat anti-rabbit IgG (H+L) were purchased from Proteintech Group (Wuhan, China). A goat anti-mouse IgG (H+L) secondary antibody was purchased from Thermo Fisher Scientific (Waltham, USA).

Cell culture

SH-SY5Y human neuroblastoma cells were purchased from Wuhan University Collection Center. The cells were cultured in DMEM-F12 medium containing a mixture of 15% fetal bovine serum and 1% penicillin streptomycin in a 5% CO2 cell culture incubator at 37 °C. When the cells covered approximately 80% of the bottom of the bottle, they were collected in a single tube and centrifuged at 1000 rpm for 5 min after trypsin digestion. After discarding the supernatant, the cells were seeded in 96-well plates, 24-well plates, or 6-well plates at a density of 5  × 104/mL in complete medium, and the medium was changed every 2 days.

Determination of the optimal multiplicity of infection (MOI)

Virus transduction construct, cloning, and production were carried out by Hanbio Biotechnology Co., Ltd. The detailed steps and more information are provided in the supplementary files. SH-SY5Y cells showing good growth were chosen. The SH-SY5Y cells (1 ×105 cells/mL) were plated in a 96-well plate at a density of 1 ×104 cells per well and maintained at 37 °C and 5% CO2 A total of 200 µL of fresh medium containing virus at an MOI of 0, 10, 20, 30, 40, 50, or 60 (including 1.2 µL of polybrene at a working concentration of 6 µg/mL). After 12 h, the transfection was terminated, and the optimal MOI was selected by observing and photographing the cells under a fluorescence microscope.

Cell viability determination

Cell viability was assessed by the CCK-8 assay (Apexbio, USA); in this assay, 2-(2-methoxy-4-nitrophenyl)-3-(4-nitrophenyl)-5-(2,4-disulfophenyl)-2H-tetrazolium monosodium salt produces water-soluble maz dye after it is formed and undergoes biological reduction in the presence of electron carriers. Briefly, SH-SY5Y cells (1 ×105 cells/mL) were seeded in each well of a 96-well plate for 24 h. After treatment, CCK-8 solution (10 µL) was added to each well of the 96-well plate, the cells were incubated at 37 °C for 2 h. The absorbance was measured at 450 nm with a Synergy HTX microplate reader (Bio Tek, USA).

Real-time fluorescent quantitative PCR (qPCR)

TransZol Up reagent was used to extract total RNA. For first-strand cDNA synthesis and gDNA removal, RNA template and primers (Table 1) were designed by Sangon Biotech Co., Ltd. (Shanghai, China), mixed with RNase-free water, incubated at 65 °C for 5 min, incubated on ice for 2 min, and then added to other reaction components (Table 2). qPCR was performed on a t100 thermal cycler/621 BR 18022 gradient PCR instrument (Bole Biotechnology Co., Ltd.) using NuHi Robustic SYBR Green Mix (Suzhou Nuhigh Biotechnologies Co., Ltd.). Each reaction (25 µL volume) consisted of 1.5 µL cDNA, 12.5 µL NuHi Robustic SYBR Green Mix, 0.5 µL forward and reverse primers (10 µM) and 10 µL RNase-free double distilled water. The qPCR conditions were as follows: initial denaturation at 95 °C for 2 min, 40–50 cycles at 95 °C for 5 s, and 55 °C for 25 s. Melting curve analysis was performed from 55 °C to 95 °C to determine the specificity of the qPCR primers.

Table 1 Sequences of carrier primers.

Name	Sequences	
GAPDH Forward Primer	GGAGCGAGATCCCTCCAAAAT	
GAPDH Reverse Primer	GGCTGTTGTCATACTTCTCATGG	
qTDP43-F1	AGGTGGCTTTGGGAATCAGG	
qTDP43-R1	CCCAACTGCTCTGTAGTGCT	
Notes.

The carrier primers were designed by Sangon Biotech Corporation, Ltd. (Shanghai, China).

Table 2 Reagents and volume.

Component	Volume	
Total RNA	50 ng–5 µg	
Anchored Oligo (dT)18 Primer (0.5 µg/µL)	1 µL	
Or Random Primer (0.1 µg/1 µL)	1 µL	
2x ES Reaction Mix	10 µL	
EasyScript RT/RI Enzyme Mix	1 µL	
gDNA Remover	1 µL	
RNase-free Water	Variable	
Total volume	20 µL	

Western blot assay

Cells were collected, lysed with RIPA lysis buffer, and then centrifuged at 12,000 rpm at 4 °C for 15 min. The total protein concentration was measured with a BCA protein assay kit (Beijing Solarbio). The proteins were then heated at 100  °C for 5 min for denaturation. Equal amounts of total protein (25 µg per lane) were loaded on 12% SDS-PAGE gels and separated. After sample loading, the voltage was set to 90 V and then to 120 V until the sample reached the bottom of the gel. The proteins were then transferred to a nitrocellulose (NC) film by the sandwich method. The membrane was washed three times with TBST and then blocked with 5% skimmed milk. The membrane was then incubated for 12 h at 4 °C with a GAPDH antibody (1:1000; Proteintech Group), TDP-43 antibody (1:1000; Proteintech Group), and CytC antibody (1:1000; Proteintech Group). The membrane was washed with TBST 3 times and incubated with HRP-conjugated Affinipure goat anti-rabbit lgG (H+L) (1:1000; Proteintech Group) for 2 h at room temperature. The membranes were developed using hydrogen peroxide and Supersignal West Pico Luminol (Pierce, Seymour Fisher Technologies). Finally, High-Sig ECL Western Blotting Substrate (Shanghai Tanon Technology Co., Ltd.) was used to visualize the membrane.

JC-1 assay

A JC-1 MMP test kit (Shanghai Yeasen Biotechnology Co., Ltd.) was used according to the manufacturer’s protocol. The culture medium of each group of cells was discarded, the cells were wash once with PBS, and the same amount of culture medium and JC-1 staining working solution was added and mixed well. Afterwards, the cells were placed in a 37 °C cell incubator and incubated for 20 min. Then, the supernatant was discarded, the cells were washed twice with prepared 1 × JC-1 staining buffer, and cell culture medium was added. After this, the cells were washed with PBS and observed under an inverted fluorescence microscope (IX73; Olympus, Japan).

Changes in ATP content, T-AOC, GSH-Px activity, total SOD (T-SOD) activity, copper/zinc SOD (CuZn-SOD) activity, manganese SOD (Mn-SOD) activity and MDA content

We used a commercial ATP content test kit, GSH-Px test kit, SOD typing test kit, MDA test kit (Nanjing Jiancheng Biological Engineering Institute) and T-AOC test kit (Shanghai Beyotime Biotechnology Co., Ltd.). A BCA protein quantitative kit (Beijing Solarbio) was used to determine the protein content.

To measure ATP levels, reagents and the sample to be tested were added to tubes according to the manufacturer’s protocol. A blank tube, a standard tube, experimental tubes, and a control tube were analyzed, and the experiment was performed in triplicate. The absorbance value of each tube at a wavelength of 636 nm was measured (Multiskan GO, Thermo Scientific, USA).

To determine T-AOC, cells were scraped off, lysed on ice, and centrifuged at 4 °C at 12,000 rpm in a low-temperature centrifuge for 5 min, and the supernatant was taken for determination. Reagents and the sample to be tested were added to each well according to the manufacturer’s instructions, the absorbance value of each well at a wavelength of 734 nm was measured, and the T-AOC of the sample was calculated according to the standard curve.

To measure GSH-Px levels, prepared cell homogenates were centrifuged in a low-temperature centrifuge at 2000 rpm for 10 min, and the supernatant was centrifuged at 10,000 rpm in a low-temperature centrifuge for 15 min. After adding 0.5 mL of PBS and mixing, the prepared detection reagents were added, and the absorbance value of each sample was determined at a wavelength of 412 nm.

To analyze SOD levels, T-SOD, CuZn-SOD, and Mn-SOD activity in SH-SY5Y cells was measured using a SOD typing assay kit via the hydroxylamine method according to the manufacturer’s protocol. SH-SY5Y cells were seeded into a 6-well plate, and the absorbance of the samples was measured at 550 nm using a microplate reader.

To assess MDA levels, treated cells were washed with cold PBS and homogenized in PBS in a glass homogenizer to form a suspension. The supernatant was used for analysis according to the manufacturer’s protocol, and the absorbance of each sample was measured at a wavelength of 530 nm.

Statistical analysis

All statistical analyses were performed with SPSS 22.0 (IBM, USA). All results are expressed as the mean ± SD. The difference between the means of more than two groups was analyzed by one-way analysis of variance (ANOVA). When ANOVA showed significant differences, pairwise comparisons of means were made by Bonferroni’s post hoc t-test with correction. A value of P < 0.05 was considered statistically significant.

Results

Establishment of a TDP-43-transfected SH-SY5Y cell model

Cell were transfected with virus an MOI of 0, 10, 20, 30, 40, 50 or 60. Cells transfected at an MOI of 0 were in good condition and showed no green fluorescence under a fluorescence microscope. Cells transfected at an MOI of 30 were in good condition, showed clearly visible dendrites, fluoresced under a fluorescence microscope, and showed high transfection efficiency. The fluorescence and transfection efficiency of cells transfected at MOIs of 10 and 20 were lower than those transfected at an MOI of 30. The cells transfected at MOIs of 40 and 50 were in a poor state; there cell bodies were crumpled and collapsed, and some of the cells were dead. Cells transfected at an MOI of 60 were blurred or absent, and a large number of cells were dead. In conclusion, the optimal MOI was determined to be 30 (Fig. 1A). The morphology of cell transfected with TDP-43 for 12, 24 and 48 h were observed, and the cells transfected for 12 h showed the most normal morphology, although some shrunken cells were observed. At 24 h, the cells were in a poor state; most of the cells were crumpled or collapsed, and some were dead. At 48 h, the cells were in a worse state, and a large number of cells were dead (Fig. 1B). After transfection for 8, 10, 12, 14 or 16 h, the cell activity of the control group and TDP-43 group was assessed, and the results showed that after transfection for 12, 14 and 16 h, the activity of the TDP-43 group was decreased compared with that of the Con group, indicating that cell activity was reduced after transfection with TDP-43 and that the damage model was successfully established. TDP-43 had strong cytotoxicity, and the cells were in a poor state at 24 h and 48 h after transfection. However, at 12 h after transfection, the cells were in a relatively healthy state, although some cells showed morphological changes and cell activity was significantly decreased. Therefore, this time point was suitable for assessing the effect of ICT. In this experiment, cells were transfected with TDP-for 12 h (Fig. 1C). The mRNA expression level of TDP-43 in the TDP-43 group was higher than that in the control group (Fig. 1D). Western blotting was used to measure the expression of transfected TDP-43 protein, and the results showed that TDP-43 protein expression in the TDP-43 group was increased compared with that in the control group (Fig. 1E). In summary, an MOI of 30 and a transfection time of 12 h were selected to establish the TDP-43-transfected cell model.

Figure 1 Establishment of the TDP-43-transfected SH-SY5Y cell model.

(A) Determination of the optimal MOI exploration. (B) Morphological observation of cells in each group 12, 24 and 48 h after transfection. (C) Activity of the cells 8, 10, 12, 14 and 16 h after transfection. The data are shown as the mean ± SD; n = 5 (aP < 0.05 vs. the control group). (D) mRNA expression of TDP-43 in SH-SY5Y cells transfected with TDP-43. The data are shown as the mean ± SD; n = 3 (aP < 0.05 vs. the control group). (E) Expression of TDP-43 in SH-SY5Y cells transfected with TDP-43. The data are shown as the mean ± SD; n = 3 (aP < 0.05 vs. the control group).

Effects of ICT at different concentrations for different durations on the activity of TDP-43-transfected SH-SY5Y cells

After modeling, ICT at concentrations of 0.0, 0.1, 1.0 and 10.0 µmol/L was added, and the cell activity of each group at 12, 24 and 48 h were determined. The results showed that optimal cell viability was obtained at a drug concentration of 1.0 µmol/L and a treatment time of 48 h. Therefore, subsequent experiments were carried out under these conditions (Fig. 2A).

Figure 2 Cell viability.

(A) Effects of ICT at different concentrations and time points on cell activity. The data are shown as the mean ± SD; n = 5 (aP < 0.05 vs. the 0.0 µmol/L ICT group, bP < 0.05 vs. the 0.1 µmol/L ICT group, cP < 0.05 vs. the 1.0 µmol/L ICT group). (B) Effects of ICT on the activity of SH-SY5Y cells transfected with TDP-43 and the expression level of TDP-43. The data are shown as the mean ± SD; n = 5 (aP < 0.05 vs. the control group).

Effects of ICT on the viability of SH-SY5Y cells transfected with TDP-43

Changes in cell activity in the control group, control + ICT group, TDP-43 group, and TDP-43 + ICT group were assessed after treatment with 1.0 µmol/L ICT for 48 h. The results showed that compared with that of the control group, the activity of the TDP-43 group was decreased, while the cell activity of the TDP-43 + ICT group was increased after ICT treatment (Fig. 2B).

Effects of ICT on TDP-43 expression in SH-SY5Y cells transfected with TDP-43

The Western blot results showed that compared with that in the Con group, the expression level of TDP-43 in the TDP-43 group was increased; however, after ICT treatment, the expression level of TDP-43 decreased (Fig. 3).

Figure 3 Influence of ICT treatment on TDP-43 expression in the control group compared with the TDP-43 group.

The data are shown as the mean ± SD; n = 3 (aP < 0.05 vs. the control group, bP < 0.05 vs. the TDP-43 group).

Effects of ICT on ATP content, the MMP and CytC expression in SH-SY5Y cells transfected with TDP-43

JC-1, which can quickly and sensitively detect MMP changes, was used a fluorescent probe in this experiment. In normal mitochondria, JC-1 aggregates in the mitochondrial matrix to form a polymer that emits strong red fluorescence. In unhealthy mitochondria, due to a decrease in the MMP, JC-1 can only exist in the cytoplasm as monomers, which produce green fluorescence. The results showed that compared with the control group and control + ICT group, the TDP-43 group showed reduced red fluorescence, enhanced green fluorescence and a decreased MMP. Green fluorescence was weakened, red fluorescence was enhanced, and the MMP was increased in the TDP-43 + ICT group compared with the TDP-43 group (Figs. 4A and 4B).

Figure 4 Effects of ICT on the MMP, ATP content and CytC expression in TDP-43-transfected SH-SY5Y cells.

(A) The MMP of SH-SY5Y cells was monitored with the fluorescent probe JC-1 (magnification: 100×, scale bar 100 µm). (B) Quantitative analysis of the MMP in the different groups. (C) Effects of ICT on ATP content in the model cells. The data are shown as the mean ± SD; n = 3 (aP < 0.05 vs. the control group). (D) Influence of ICT on CytC expression in the model cells. The data are shown as the mean ± SD; n = 3 (aP< 0.05 vs. the control group, bP < 0.05 vs. the TDP-43 group).

The results showed that there was no significant difference in ATP content between the control + ICT group and control group and that ATP content in the TDP-43 group was lower than that in the control group. Compared with that in the TDP-43 group, the ATP content in the TDP-43 + ICT group was increased (Fig. 4C).

The Western blot results showed that compared with that in the control group, the expression level of CytC in the TDP-43 group was increased, while the expression level of CytC decreased after ICT treatment (Fig. 4D).

The effects of ICT on T-AOC, GSH-Px activity, T-SOD activity, CuZn-SOD activity, Mn-SOD activity and MDA content in TDP-43-transfected SH-SY5Y cells

T-AOC (Fig. 5A), GSH-Px activity (Fig. 5B), T-SOD activity (Fig. 5C), CuZn-SOD activity (Fig. 5D) and Mn-SOD activity (Fig. 5E) were reduced in SH-SY5Y cells transfected with TDP-43 compared with control cells. After ICT treatment, T-AOC (Fig. 5A), GSH-Px activity (Fig. 5B), T-SOD activity (Fig. 5C), CuZn-SOD activity (Fig. 5D) and Mn-SOD activity (Fig. 5E) were increased. MDA content was increased in TDP-43-transfected cells compared with control cells (Fig. 5F) and decreased after ICT treatment (Fig. 5F).

Figure 5 The influence of ICT on T-AOC, GSH-Px activity, T-SOD activity, CuZn-SOD activity, Mn-SOD activity and MDA content in model cells.

The data are shown as the mean ± SD; n = 3 (aP < 0.05 vs. the control group, bP < 0.05 vs. the TDP-43 group).

Discussion

TDP-43 is a highly conserved nuclear RNA/DNA-binding protein involved in the regulation of RNA processing (Jo et al., 2020). TDP-43 was discovered in 1995. A large number of studies have shown that in addition to having normal physiological functions, TDP-43 is closely related to the occurrence and development of neurodegeneration, possibly by abnormally accumulating, which can lead to mitochondrial damage and oxidative stress (Josephs et al., 2014; Josephs et al., 2016; LaClair et al., 2016). Under pathological conditions, TDP-43 can have toxic effects on nerve cells (Gao et al., 2019). Studies have shown that TDP-43 can be associated with neurodegenerative diseases such as ALS and FTLD (Buratti, 2018). Therefore, in this study, we established a TDP-43-overexpressing model by transfecting cells with TDP-43. According to preliminary results, we transfected SH-SY5Y cells with a TDP-43-expressing virus at an MOI of 30 for 12 h for subsequent experiments. We found that ICT significantly improved cell viability and reduced TDP-43 protein levels.

Abnormal accumulation of TDP-43 in mitochondria is an important cause of mitochondrial dysfunction and neurodegenerative diseases (Wang et al., 2019). Preventing TDP-43 from localizing in mitochondria can alleviate mitochondrial abnormalities, neuronal loss and behavioral defects in TDP-43 transgenic mice (Wang et al., 2016). Although TDP-43 levels in mitochondria were not specifically measured in this experiment, the effect of ICT on the mitochondria of TDP-43-transfected SH-SY5Y cells was observed by measuring ATP content, the MMP and CytC expression. ATP content can directly reflect the extent of mitochondrial function (Rossi et al., 2020). The MMP is formed through a concentration gradient of protons or ions on both sides of the inner membrane, which is the basic premise for normal mitochondrial physiological function (Rossi et al., 2020). Under physiological conditions, CytC rarely penetrates the outer membrane, but when the MMP decreases, the permeability of the mitochondrial membrane is increased, and CytC can be released from the mitochondrial inner membrane into the cytoplasm (Xiao et al., 2018). This study showed that ICT increased ATP content and the MMP while decreasing CytC levels in SH-SY5Y cells transfected with TDP-43. This indicates that the decrease in the MMP and release of CytC caused by the overexpression of TDP-43 leads to a decrease in ATP production and that ICT can alleviate this damage by reducing the level of TDP-43. Interestingly, some studies have proposed that a higher level of TDP-43 in mitochondria leads to a decrease in the synthesis of ATP and that this decrease in ATP synthesis leads to decreased degradation of TDP-43 by the mitochondrial protease LonP1 because this process is ATP dependent (Ruan et al., 2017; Wang et al., 2019). Therefore, we believe that ICT may partially degrade TDP-43 by disrupting this vicious cycle, but this merely a speculation because we did not assess LonP1 levels.

A large amount of TDP-43 is localized in mitochondria mainly because a long-term increase in oxidative stress increases the nuclear depletion and cytoplasmic accumulation of TDP-43 and promotes the entry of a large amount of this protein into mitochondria. The interaction between TDP-43 and OXPHOS-related mt-mRNA affects the mitochondrial respiratory chain, resulting in a decrease in ATP production, a decrease in the MMP and an increase in ROS levels. This increase in ROS levels promotes the TDP-43/OXPHOS-related mt-mRNA interaction, forming a vicious cycle (Lucini & Braun, 2021). Lipid peroxides produced by the body during oxidative stress leads to mitochondrial dysfunction by inhibiting the mitochondrial respiratory chain and enzyme activity, the extent of which can be reflected by the MDA content. There are many kinds of antioxidants in the body, including antioxidant macromolecules, antioxidant small molecules and enzymes. T-AOC refers to the overall ability of the body to resist oxidative stress. The enzymatic system includes antioxidant enzymes in the body, such as GSH-Px and SOD (Li et al., 2020a). These enzymes maintain the balance between oxidation and antioxidation by scavenging free radicals and exerting antilipid oxidation effects and play an important role in maintaining the normal function of cells and organisms. Therefore, we measured the levels of these indicators that are closely related to oxidation and antioxidation. This study showed that ICT increased T-AOC, GSH-Px activity and SOD activity while decreasing MDA levels in SH-SY5Y cells transfected with TDP-43. This indicates that overexpression of TDP-43 leads to an imbalance between oxidation and antioxidation, which can be improved by ICT.

How does ICT reduce the level of TDP-43? According to our findings and previous results, we speculated that ICT promotes the degradation of TDP-43 by increasing the levels of LonP1 and ATP. We also speculate that the mechanism is related to the antioxidant capacity of ICT based on existing experiments and related research. A large number of studies have shown that ICT is a relatively good plant-based antioxidant and it has good antioxidant capacity in different cell and animal experiments (Liu et al., 2014; Xu et al., 2021). This experiment also showed that ICT can improve the levels of oxidative stress indicators to varying degrees. This direct or indirect resistance to oxidative stress prevents the vicious cycle of abnormal TDP-43 accumulation to a certain extent. In addition, the consequential abnormal signaling of TDP-43 promotes the formation of stress granules and impairs pathways that are important for cellular degradation, such as autophagy. ICT also has a certain regulatory effect on autophagy (Yu et al., 2020); thus, autophagy may also contribute to the reduction in TDP-43 levels by ICT. According to the experimental results, the level of TDP-43 did not change significantly in the control + ICT group; thus, it can be concluded that the change in the level of TDP-43 was mainly due to the impact of exogenous TDP-43. ICT can improve mitochondrial function and alleviate oxidative stress, but there have been no reports on the effect of ICT on expression vectors. Therefore, we believe that this change in exogenous TDP-43 expression is not the result of the expression vector based on what we previously learned. Mentioned mitochondria and antioxidants. However, these hypotheses are preliminary, and further experiments are needed for verification.

It is well known that TDP-43 gene mutations are one of the recognized genetic causes of ALS, which is closely related to an imbalance of energy homeostasis and disease susceptibility and progression (Floare & Allen, 2020). Other risk factors for ALS, such as SOD1, are also intricately related to TDP-43. However, the pathogenesis of ALS is multifaceted, and it is difficult to say how TDP-43 differs from other related genetic risk factors for ALS (such as SOD1, FUS and C9orf72) or which of these risk factors is the dominant one (Allen et al., 2019; Richardson et al., 2013). Therefore, the effects of drug candidates for ALS the target TDP-43 are complex and multifaceted. In this experiment, ICT reduced the TDP-43 level in TDP-43-transfected SH-SY5Y cells, improved mitochondrial function and resisted oxidative stress, which indicates that it is a potential TDP-43-targeting drug for ALS. However, these studies are very preliminary, and more in-depth research is needed. Further clarification of the role of TDP-43 in ALS and other neurodegenerative diseases will provide data for the development of related drugs.

Conclusions

The results suggests that the protective effect of ICT against TDP-induced SH-SY5Y cell injury is related to a reduction in TDP-43 expression, alleviation of mitochondrial injury and amelioration of oxidative stress.

Supplemental Information

Supplemental Information 1 Raw data of Figs. 1–5

Click here for additional data file.

Supplemental Information 2 Full length uncropped blots of Figs. 1, 3 and 4

Click here for additional data file.

Supplemental Information 3 Virus transduction construct, cloning, and production

Click here for additional data file.

Additional Information and Declarations

Competing Interests

Author Contributions

Data Availability

The authors declare there are no competing interests.

Yongjian Zhou and Nanqu Huang conceived and designed the experiments, performed the experiments, analyzed the data, prepared figures and/or tables, authored or reviewed drafts of the paper, and approved the final draft.

Yuanyuan Li conceived and designed the experiments, performed the experiments, authored or reviewed drafts of the paper, and approved the final draft.

Zhisheng Ba, Yanjun Zhou and Yong Luo conceived and designed the experiments, authored or reviewed drafts of the paper, and approved the final draft.

The following information was supplied regarding data availability:

The raw measurements are available in the Supplemental Files.

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
