# Peer review of "Icaritin protects SH-SY5Y cells transfected with TDP-43 by alleviating mitochondrial damage and oxidative stress"

_PeerJ, doi:10.7717/peerj.11978_

## Round 0.1 · original submission · Major Revisions

Dear authors:
Thank you for submitting your manuscript to PeerJ. Your manuscript has now been reviewed by two experts in the field. Both reviewers agree that the manuscript requires major revisions. I agree with the reviewers' comments and their suggestions for improvement.

As you can see from their remarks, the reviewers have raised specific points for all sections of the manuscript, and they suggest how the manuscript has to be improved. The revisions include, but are not limited to, the appropriate description of experimental methods and sufficient details on the statistical evaluation of your data. Furthermore, the reviewers point out essential control experiments that are missing.
With respect to the presentation of your data, please include the full-size original blots in the Supplemental Data file and add the position of molecular weight markers. Also, provide evidence that the signals for Western blotting are not saturated and in the linear range. Please follow the suggestions of Reviewer 1 for the presentation of data and statistical evaluations. As discussed by Reviewer 2, better quality high-resolution images are required, and quantitative analyses of MMP changes are recommended.

Collectively, both reviewers pinpoint the additional experiments and considerable editing that are mandatory at this stage. I hope that you will be able to use the constructive comments made by the reviewers to improve the experimental work, data presentation, and writing for a re-submission.

I encourage you to address and respond to each of the reviewers’ comments before re-submitting your revised manuscript. Once the revised manuscript has been received, it will undergo a second round of review. I am looking forward to receiving your revised manuscript in the near future.

·

Basic reporting

Using viral transduction, the authors have overexpressed TDP43 in SY5Y cells, and then tried to recover the defects observed using Icaritin. This approach has been used previously to target mitochondrial function.

Overall, the hypothesis is sound and the data in general looks robust. However, the manuscript suffers from a chronic lack of detail, especially in the methodology section. Which is quite poor in terms of content, scientific clarity and presentation (constant switching between current and past tense, incorrect use of English). There also needs to be more clarity on the statistical methods used.

For the methods, there is little description in the figure legends, it is not clear what stats were used, the data should not be presented in block bars, individual data point should be shown and standard deviation not standard error should be used. The westerns contain no molecular weight markers and full length blots should be shown in the Supplementary data.

The conclusions are essentially a description of the results without historical literature contextualization and no discussion on what the results could mean in the context of ALS.

I suggest major revisions before being accepted.

Experimental design

The approach is valid but the lack of methodological detail is not acceptable especially around the use of the TDP virus the activity assays in figure 5 and the statistical approach.

Validity of the findings

The data need to be presented in a more transparent fashion, removing the use of block histograms and using individual data points, standard deviation, describing in detail the statistical approach used in every figure legend and putting markers on blots. Full length blots should be provided in the Supplementary section

Additional comments

Introduction
Please add more up to date TDP43 articles focusing on TDP43 mitochondrial dysfunction.
Correct numerous grammatical errors.
Re-write lines 49-52, they dont read well.

Methods- Please stick with past tense and correct numerous grammatical errors and use of incorrect English.
Please add detail on the following
Cell culture
Virus transduction-construct, cloning, production, virus titration,
Western blot- 50g per lane were added? I assume this is a mistake, please amend. Pleas provide evidence that all blots were in the linear range. The GAPDH loading controls look saturated.
Add full detailed procedures for ATP/GSH/SOD/MDA/TAOC/JC-1
Stats- please add what normality testing was performed and what stats were performed for each assay. put this in the figure legends. For example most of the Figure Legends state stats were control vs TDP43 group or TDP vs Icaritin group which suggest two groups were compared only. But the methods state annova was performed which is performed on three or more groups. Please amend

Results- please present all data showing individual data points and standard deviation not standard error, please add descriptive text to westerns blots molecular weight, which bands represent TDP43. Figure 1/3 for example shows two TDP43 bands, what do each band represent. Please put full length westerns in the Supplementary data.
Figure 4B, please provide quantification of the JC-1 data
Figure 4C-the quantification does not match the picture

Discussion, please provide detailed contextualized discussion of the data in terms of the existing TDP43 literature and what these results mean in the context of TDP43 diseases such as ALS. Listing the results is not a discussion.

Reviewer 2 ·

Basic reporting

The rationale of this study is straight forward, and the suppression of TDP-43 toxicity is an area of interest. However, data quality, data presentation and method details in several places need significant improvements to strengthen the manuscript. In addition to experimental results, original research articles are not sufficiently cited in the introduction. It is also strongly recommended that the manuscript should be extensively edited and proofread by authors and professional editors.

Experimental design

1. (Figure 3) One key observation of this article is the reduction of TDP-43 after ICT treatment.
1a. Authors state that TDP-43 is significantly reduced in the TDP-43+ICT group. However, high quality western blot should be provided to support this conclusion. In current western blot, TDP-43 bands in the TDP-43 +ICT lane are fractured, which could lead to signal loss and underestimation of protein level. Also, the background signal of the blot is high. I would suggest the authors to optimize western blot procedures (e.g. antibody concentration, washing condition, amount of protein loaded, film exposure time) before performing bands quantification.

1b. More information about figure 3 data should be provided in the main text. It is unclear whether the exogenous, endogenous or both TDP-43 expression were reduced after ICT treatment? If the endogenous TDP-43 was reduced in TDP-43 + ICT, why did the endogenous TDP-43 level remain unchanged in Con + ICT? If only the exogenous TDP-43 was affected, authors should discuss whether ICT has any possible effects on the expression vector.

1c. Since TDP-43 repression by ICT is a main point of this article, readers would expect a bit more characterizations or in-depth discussion on how ICT could achieve it. For example, could ICT promote TDP-43 degradation through autophagy or ubiquitin-proteosome system?

2. (Figure 4B) Authors observed reduced MMP after TDP-43 transfection, and ICT treatment prevented loss of MMP in TDP-43-transfected cells.
2a. Current image resolution is too low, and it is difficult to evaluate authors’ conclusion regarding changes in JC-1 color in different groups. To support authors’ conclusion, high resolution microscopic images should be provided.

2b. To strengthen the point that ICT protects cells from TDP-43-induced MMP loss, quantitative analysis showing MMP changes across treatment conditions is highly recommended. Specifically, the red/green ratio of JC-1 in each experimental condition should be presented.

2c. It would be of authors’ interest to include mitochondrial uncoupler treatment such as CCCP as a positive control in the JC-1 assay.

2d. Line 185: JC-1 can only exist in the cytoplasm as “monomers”, not as a “monocyte”.

3. (Supplementary file 2) The corresponding films/bands of main figure 1E, 3 and 4C are not indicated in the uncropped western blots. I recommend that information such as protein names, experimental conditions and connections of these films to main figures should be included.

4. (Materials and Methods) Some methods and experimental details need clarification.
Line 58: Please provide human TDP-43 vector information.
Line 104: I think “50 g” of protein loaded to each lane is a typo, please fix.
Line 108: Please provide the source and dilution of GAPDH, TDP-43 and cytochrome c antibodies.

Validity of the findings

Data quality improvement and more controls are required to support the main conclusion that ICT suppresses TDP-43 toxicity by reducing its level.

Additional comments

1. (Introduction) Basis of the hypothesis in the introduction needs to be clarified.
Line 46: The original research articles about Icaritin improving structure and function of brain mitochondria in transgenic AD mouse should be cited. The authors cite a review paper (Angeloni et al. 2019) about Icariin, but I couldn’t find the references in this review related to Icaritin improving mitochondrial functions in AD mouse. Please cite the original research here.
Line 48: The authors propose that TDP-43 reduction is a possible mechanism for ICT to improve mitochondrial functions, but this hypothesis seems irrelevant to the context of background. Relevant references should be cited in the introduction to elaborate why authors propose this hypothesis. For example, any evidence that show ICT could promote degradation of misfolded proteins?

2. (Discussion)
Line 201: Please clarify “TDP-43 is a highly conserved ribosome”. I believe this is not the current knowledge about ribosome and TDP-43.

---

## Round 0.2 · Minor Revisions

Dear Authors:

Thank you for submitting your revised manuscript. As you can see from the reviewers’ comments, the revisions have markedly improved the manuscript.

However, concerns remain about the Western blot data. I fully agree with the comments previously made by Reviewer 1: “Western blot- 50g per lane were added? I assume this is a mistake, please amend. Please provide evidence that all blots were in the linear range. The GAPDH loading controls look saturated.”

The Western blots in Fig. 3 and Fig. 4C are of low quality and bands appear saturated. Please replace these blots with higher-quality data and show that signals are not saturated.

In light of these comments, the decision on the manuscript is “Minor revisions”.

I am looking forward to receiving your revised manuscript!

·

Basic reporting

The authors in the most part have addressed all my concerns

Experimental design

The authors in the most part have addressed all my concerns. My remaining concern would be the western data with many of the bands looking saturated.

Validity of the findings

The authors in the most part have addressed all my concerns.

Additional comments

The authors in the most part have addressed all my concerns. My remaining concern would be the western data with many of the bands looking saturated.

Reviewer 2 ·

Basic reporting

The manuscript has been improved significantly with much better writing, data quality and presentation. The authors have also addressed most of the issues I raised.

Experimental design

no comment

Validity of the findings

no comment

Additional comments

Thank you for your response to my comments.

---

## Round 0.3 · accepted · Accept

Dear Authors:

Thank you for submitting your interesting work to PeerJ.

With best regards - Ursula Stochaj